# Revisiting Model Stitching to Compare Neural Representations

**Yamini Bansal**
Harvard University
ybansal@g.harvard.edu

**Preetum Nakkiran**
Harvard University
preetum@cs.harvard.edu

**Boaz Barak**
Harvard University
b@boazbarak.org

## Abstract

We revisit and extend *model stitching* (Lenc & Vedaldi 2015) as a methodology to study the internal representations of neural networks. Given two trained and frozen models $A$ and $B$, we consider a "stitched model" formed by connecting the bottom-layers of $A$ to the top-layers of $B$, with a simple trainable layer between them. We argue that model stitching is a powerful and perhaps under-appreciated tool, which reveals aspects of representations that measures such as centered kernel alignment (CKA) cannot. Through extensive experiments, we use model stitching to obtain quantitative verifications for intuitive statements such as "good networks learn similar representations", by demonstrating that good networks of the same architecture, but trained in very different ways (e.g.: supervised vs. self-supervised learning), can be stitched to each other without drop in performance. We also give evidence for the intuition that "more is better" by showing that representations learnt with (1) more data, (2) bigger width, or (3) more training time can be "plugged in" to weaker models to improve performance. Finally, our experiments reveal a new structural property of SGD which we call "stitching connectivity", akin to mode-connectivity: typical minima reached by SGD can all be stitched to each other with minimal change in accuracy.

## 1 Introduction

The success of deep neural networks can, arguably, be attributed to the intermediate *features* or *representations* learnt by them [Rumelhart et al., 1985]. While neural networks are trained in an end-to-end fashion with no explicit constraints on their intermediate representations, there is a body of evidence that suggests that they learn rich a representation of the data along the way [Goh et al., 2021, Olah et al., 2017]. However, theoretically we understand very little about how to formally characterize these representations, let alone *why* representation learning occurs. For instance, there are various ad-hoc pretraining methods [Chen et al., 2020a,b], that are purported to perform well by learning good representations, but it is unclear which aspects of these methods (objective, training algorithm, architecture) are crucial for representation learning and if these methods learn qualitatively different representations at all.

Moreover, we have an incomplete understanding of the *relations* between different representations. Are all "good representations" essentially the same, or is each representation "good" in its own unique way? That is, even when we train good end-to-end models (i.e., small test loss), the internals of these models could potentially be very different from one another. A priori, the training process could evolve in either one of the following extreme scenarios (see Figure 1):

**(1)** In the "snowflakes" scenario, training with different initialization, architectures, and objectives (e.g., supervised vs self-supervised) will result in networks with very different internals, which are completely incompatible with one another. For example, even if we train two models with identical data, architecture, and task, but starting from two different initializations, we may end up at local

35th Conference on Neural Information Processing Systems (NeurIPS 2021).

minima with very different properties (e.g., Liu et al. [2020]). If models are trained with different data (e.g., different samples), different architecture (e.g., different width), or different task (e.g., self-supervised vs supervised) then they could end up being even more different from one another.

**(2)** In the "Anna Karenina" scenario[1], all successful models end up learning roughly the same internal representations. For example, all models for vision tasks will have internal representation corresponding to curve detectors, and models that are better (for example, trained on more data, are bigger, or trained for more time) will have better curve detectors.

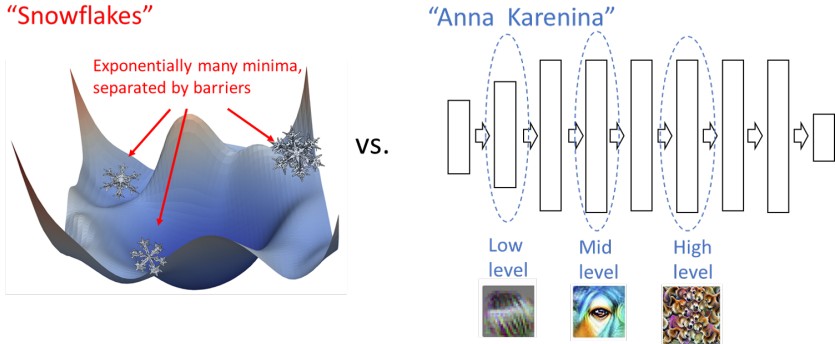

Figure 1: Two extreme "cartoons" for training dynamics of neural networks. In the "snowflakes" scenario, there are exponentially many well-performing neural networks with highly diverging internals. In the "Anna Karenina" scenario all well-performing networks end up learning similar representations, even if their initialization, architecture, data, and objectives differ. Image credits: Li et al. [2018], Olah et al. [2017, 2020], Komarechka [2021].

The "Anna Karenina" scenario implies the following predictions:

*"All roads lead to Rome:"* Successful models learned with different initializations, architectures, and tasks, should learn similar internal representations, and so if $A$ and $B$ are two such models, it should be possible to "plug in" the internals from $A$ into $B$ without a significant loss in performance. See Figure 2A-B.

*"More is better:"* Better models trained using more data, bigger size, or more compute, should learn better versions of the same internal representations. Hence if $A$ is a more successful model than $B$, it should be possible to "plug in" $A$'s internals to $B$ and obtain *improved performance*. See Figure 2C.

In this work, we revisit the empirical methodology of "model stitching" to test the above predictions. Initially proposed by Lenc and Vedaldi [2015], model stitching is natural way of "plugging in" the bottom layers of one network into the top layers of another network, thus forming a stitched network (however care must be taken in the way it is performed, see Section 2). We show that model stitching has some unique advantages that make it more suitable for studying representations than representational similarity measures such as CKA [Kornblith et al., 2019] and SVCCA [Raghu et al., 2017]. Our work provides quantitative evidence for the intuition, shared by many practitioners, that the internals of neural networks often end up being very similar in a certain sense, even when they are trained under different settings.

## 1.1 Summary of Results

**Model stitching as an experimental tool.** We establish *model stitching* as a way of studying the representations of neural networks. A version of model stitching was proposed in Lenc and Vedaldi [2015] to study the equivalence between representations. In this work, we argue that the idea behind model stitching is more powerful than has been appreciated: we analyze the benefits of model-stitching over other methods to study representations, and we then use model-stitching to establish an number of intuitive properties, including new results on the properties of SGD.

---

[1]"All happy families are alike; each unhappy family is unhappy in its own way." Leo Tolstoy, 1877. https://en.wikipedia.org/wiki/Anna_Karenina_principle

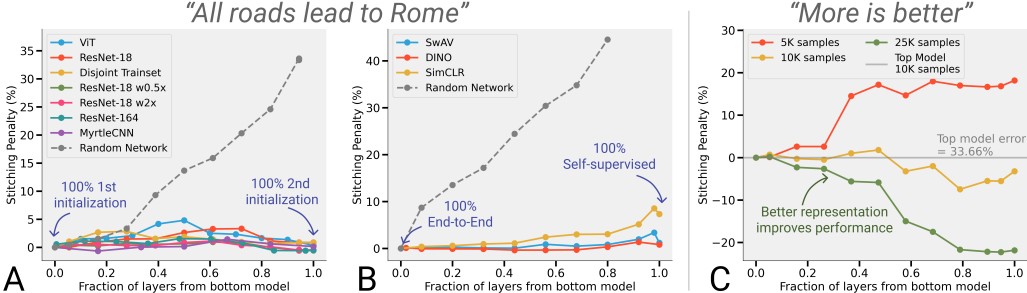

Figure 2: **Summary of main results** (A) Various models trained on CIFAR-10 identically except with different random initializations are "stitching connected": can be stitched at all layers with minimal performance drop (see Section 4). Stitching with a random bottom network shown for reference. (B) Models of the same architecture and similar test error, but trained on ImageNet with end-to-end supervised learning versus self-supervised learning can be stitched with good performance (see Section 5). (C) Better representation obtained by training the network with more samples can be "plugged-in" with stitching to improve performance (see Section 6). In all figures, stitching penalty is the difference in error between the stitched model and the base top model.

In this paper, we use model stitching in the following way. Suppose we have a neural network $A$ (which we'll think of as the "top model") for some task with loss function $\mathcal{L}$ (e.g. the CIFAR-10 or ImageNet test error). Let $r : \mathcal{X} \to \mathbb{R}^d$ be a candidate "representation" function, which can come from the first (bottom-most) layers of some "bottom model" $B$. Our intuition is that *$r$ has better quality than the first $\ell$ layers of network $A$ if "swapping out" these layers with $r$ will improve performance.*

"Swapping out" is performed by introducing an additional trainable stitching layer to $r$ with $A$ (defined more formally in Section 2). The stitching layers have very low capacity, and are only meant to "align" representations, rather than improving the model.

**Comparison to Representational Similarity Metrics.** Much of the current work studying representations focuses on similarity metrics such as CKA. However, we argue that model stitching can be a better suited tool to study representations in various scenarios, and can give qualitatively different conclusions about the behavior of neural representations compared to these metrics. We analyze the differences between model stitching and prior similarity metrics in Section 3.

**Quantitative evidence for intuitions.** Using model stitching, we are able to provide formal and quantitative evidence to the intuitions mentioned above. In particular we give evidence for the *"all roads lead to rome"* intuition by showing compatibility of networks with representation that are trained using **(1)** different initializations, **(2)** different subsets of the dataset, **(3)** different tasks (e.g., self-supervised or coarse labels). See Figure 2A-B for results. We also show that network with different random initialization enjoy a property which we call *stitching connectivity*, wherein almost all minima reachable via SGD can be "stitched" to each other with minimal loss of accuracy. We also give evidence for the *"more is better"* intuition by showing that we can *improve* the performance of a network $A$ by plugging in a representation $r$ that was trained with **(1)** more data, **(2)** larger width, or **(3)** more training epochs. See example with more samples in Figure 2C.

The results above are not surprising, in the sense that they confirm intuitions that practitioners might already have. However model stitching allows us to obtain quantitative and formal measures of these in a way that is not achievable by prior representation measures.

**Organization.** We first define model stitching formally in Section 2. We compare it with prior work on representational similarity measures in Section 3. Then, we formally define stitching connectivity and provide experimental evidence for it in Section 4. Finally, in Section 5 and 6, we provide quantitative evidence for the "all roads lead to Rome" and "more is better" intuitions respectively.

**Related works.** As mentioned above, stitching was introduced by Lenc and Vedaldi [2015] who studied equivalence of representations. They showed that certain early layers in a network trained on the Places dataset [Zhou et al., 2014] are compatible with AlexNet (see Table 4 in Lenc and Vedaldi [2015]). After completing this work, we were made aware of concurrent work Csiszárik et al. [2021] that also proposes model stitching to compare neural representations. The results from this

work complement ours by studying the effect of changing the stitching layer in various ways (for instance, imposing a sparsity penalty on the stitching layer), and further clarifying the relationship of stitching with other representational similarity measures. In contrast, our work demonstrates stitching compatibility under different scenarios, like the comparison between supervised and self-supervised methods and experiments that show "more is better".

Our work on stitching connectivity is related to the work on *mode connectivity* [Garipov et al., 2018, Freeman and Bruna, 2017, Draxler et al., 2018]. These works show that the local minima found by SGD are often connected through low-loss paths. These paths are generally non-linear, though it was shown that there are linear paths between these minima if they are identically initialized but then use different SGD noise (order of samples) after a certain point in training [Frankle et al., 2020]. Stitching connectivity is complementary to mode connectivity. Stitching connectivity corresponds to a discrete path (with as many steps as layers) but one where the intermediate steps are interpretable. We also show stitching connectivity of networks that are trained on different tasks. Finally, most of the prior work on concrete metrics for relating representations was in the context of *representation similarity measures*. We describe this work and compare it to ours in Section 3.

## 2 Model Stitching

Let $A$ be a neural network of some architecture $\mathcal{A}$, and let $r : \mathcal{X} \to \mathbb{R}^d$ be a candidate "representation" function. We consider a family $\mathcal{S}$ of *stitching layers* which are simple (e.g. linear $1 \times 1$ convolutional layers for a convolutional network $A$) functions mapping $\mathbb{R}^d$ to $\mathbb{R}^{d_\ell}$ where $d_\ell$ is the width of $A$'s $\ell$-th layer. Given some loss function $\mathcal{L}$ (e.g., CIFAR-10 or ImageNet test accuracy) we define

$$\mathcal{L}_\ell(r; A) = \inf_{s \in \mathcal{S}} \mathcal{L}(A_{>\ell} \circ s \circ r) \tag{1}$$

where $A_{>\ell}$ denotes the function mapping the activations of $A$'s $\ell$th layer to the final output, and $\circ$ denotes function composition. That is, $\mathcal{L}_\ell(r; A)$ is the smallest loss obtained by stitching $r$ into all but the first $\ell$ layers of $A$ using a stitching layer from $\mathcal{S}$. We define the *stitching penalty* of the representation $r$ with respect to $A$ (as well as $\ell$ and the loss $\mathcal{L}$) as $\mathcal{L}_\ell(r; A) - \mathcal{L}(A)$. If the penalty is non-negative, then we say that the representation $r$ is at least as good as the first $\ell$ layers of $A$.

Consider the simple case of a fully-connected network and a stitching family of linear functions. In this case, model stitching tells us if there is a way to linearly transform the representation $r$ into that of the first $\ell$ layers of $A$, but *only* in the subspace that is *relevant* to the achieving low loss. In practice, we approximate the infimum in Equation 1 by using gradient methods— concretely, by optimizing the stitching-layer using the train set of the task $\mathcal{L}$. The stitching penalty itself is then estimated on the test set.

**Choosing the stitching family $\mathcal{S}$:** The stitching family can be chosen flexibly depending on the desired invariances between representations, as long as it is simple (see below). For instance, if we choose $S$ to be all permutations or orthogonal matrices, the stitching penalty will be invariant up to permutations or orthogonal transformations respectively. In this work, we mainly consider cases where $r = B_{<l}$ for $B$ with architecture similar to $\mathcal{A}$. We restrict the stitching family per architecture $\mathcal{A}$ such that the composed model $A_{>\ell} \circ s \circ r$ lies in $\mathcal{A}$. This way the stitched model consists of layers identical to the layers of either $A$ or $B$, with the exception of just one layer. For example, for convolutional networks we consider a $1 \times 1$ convolution and for transformers we consider a token-wise linear function between transformer blocks. We perform an ablation with kernels of different sizes in Appendix B.1.

**Stitching is not learning:** One concern that may arise is that if the stitching family $\mathcal{S}$ is sufficiently powerful, it can *learn* to transform any representation into any other representation. This would defeat the aim of faithfully studying the original representations. To avoid this, the family of stitchers should be chosen to be simple (e.g., linear). Nevertheless, to verify that our experiments are not in such a regime, we stitch an untrained, randomly initialized network with a fully trained network. Figure 2A-B shows that the penalty of such a stitched network is high, specially for high $l$. CKA has the same trend for the same networks (See Appendix B.2)

**Model stitching as a "happy middle".** Model stitching can be considered as a compromise between the following extremes: **(1)** *Direct plugging in:* the most naive interpretation of "plugging in" $r$ into $A$ would be to use no stitching at all. However, even in cases where networks are identical up to a permutation of neurons, plugging in $r$ into $A$ would not work. **(2)** *Full fine tuning:* the other extreme

interpretation is to perform full *fine tuning*. That is, start with the initialized network $A_{>\ell} \circ r$ and optimize over all choices of $A$. The problem with this approach is that there are so many degrees of freedom in the choices for $A_{>\ell}$ that the resulting network could achieve strong performance regardless of the quality of $r$. For example, in B.3 we show that fine tuning can fail to distinguish between a trained network and a random network. **(3)** *Linear probe:* a popular way to define quality of representations is to use *linear probes* [Alain and Bengio, 2016]. However, linear probes are not as well suited for studying the representation of early layers, which have low linear separability. In particular, the linear probe accuracy of an early layer of network $A$ would generally always be much worse than a later layer of network $B$, even if $A$ was of "higher quality" (e.g., trained with more data). Linear probes also don't have the operational interpretation of compatibility.

By choosing a trainable but low-capacity layer, model stitching "threads the needle" between simply plugging in, and full fine tuning, and unlike linear probes, enables the study of early layers using powerful nonlinear decoders.

**Experimental setup.** Unless specified otherwise, the CIFAR-10 experiments are conducted on the ResNet-18 architecture (with first layer width 64) and the ImageNet experiments are conducted on the ResNet-50 architecture [He et al., 2015]. The ResNets are trained with the standard hyperparameters (See Appendix A.1 for training parameters of all base models). The stitching layer in the the the convolutional networks consist of a $1 \times 1$ convolutional layer with input features equal to the number of channels in $r$, and output features equal to the output channels of $A_{\leq l}(x)$. We add a BatchNorm (BN) layer before and after this convolutional layer. Note that the BN layer does not change the representation capacity of the stitching layer and only aides with optimization. We perform stitching only between ResNet blocks (and not inside a block), but note that it is possible to stitch within the block as well. We use the Adam optimizer with the cosine learning rate decay and an initial learning rate of $0.001$. Full experimental details for each experiment are described in Appendix A.

## 3   Stitching vs. representational similarity

Much prior work studying representations focused on *representation similarity measures*. These have been studied in both the neuroscience and machine learning communities [Kriegeskorte et al., 2008, Kornblith et al., 2019]. Examples of such measures include *canonical correlation analysis (CCA)* [Hardoon et al., 2004] and its singular-vector and projection-weighted variants such as SVCCA and PWCCA [Raghu et al., 2017, Morcos et al., 2018]. Recently Kornblith, Norouzi, Lee, and Hinton [2019] proposed *centered kernel alignment (CKA)* that addressed several issues with CCA. CKA was further explored by Nguyen et al. [2021].

For two representations functions $\phi : \mathcal{X} \to \mathbb{R}^{d_1}$ and $\sigma : \mathcal{X} \to \mathbb{R}^{d_2}$, the linear CKA is defined as $\mathrm{CKA}(\phi, \sigma) := \frac{||\mathrm{Cov}(\phi(x), \sigma(x))||_F^2}{||\mathrm{Cov}(\phi(x))||_F \cdot ||\mathrm{Cov}(\sigma(x))||_F}$ where all covariances are with respect to the test distribution on inputs $x \sim \mathcal{D}$ [Kornblith et al., 2019].

| Method | *Stitching Connectivity* Different Initialization | *"All roads"* Self-Supervision | *"More is better"* More Data / Time / Width |
|---|---|---|---|
| CKA | Varies (can be 0) | Varies ($0.35 - 0.9$) | Far (can be 0 for data, 0.7 for width) |
| **Stitching** | **Close** (up to $3\%$ error) | **Close** (up to $5\%$ error) | **Better** |

Table 1: Qualitative results of our experiments, comparing CKA to stitching. CKA $\approx 1$ means representations are close according to CKA. Error $\approx 0\%$ means representations are close according to stitching. "Varies" means no consistent conclusion across different architectures and tasks.

Table 1 contains a qualitative summary of our results, comparing stitching with CKA. While in some cases the results of stitching and CKA agree, in several cases, stitching obtains results that align more closely with the intuitions that well-performing networks learn similar representations, and that more resources results in better versions of the same representations. In particular, in experiments where stitching indicates that a certain representation is *better* than another, CKA by its design can only indicate that the two representations are far from each other. Moreover, in some experiments CKA indicates that representations are far where we intuitively believe that they should be close, e.g. when networks only differ by two random initializations, or when one is trained with a supervised and

another with a self-supervised task.[2] In contrast, in these settings, model-stitching reveals that the two models have nearly equivalent representations, in the sense that they can be stitched to each other with low penalty. The precise experimental results appear in Sections 4-6 and Appendix B.2.

Compared to representation-similarity measures, model stitching has several advantages:

*Ignoring spurious features:* Measures such as CCA and CKA ultimately boil down to distances between the feature vectors, but they do not distinguish between features that are learned and relevant for downstream tasks, and spurious features, that may be completely useless or even random. For example, suppose we augment a representation $\phi : \mathcal{X} \to \mathbb{R}^d$ by concatenating 1000 "useless" coordinates, with random gaussian features, to form a new representation $\phi' : \mathcal{X} \to \mathbb{R}^{d+1000}$. This would reduce the CKA, but representation $\phi'$ is not different from $\phi$ in a meaningful way. Model-stitching resolves this, since we can stitch $\phi'$ in place of $\phi$ by simply throwing away the useless coordinates. In general, model-stitching focuses only on aspects of the representation which are relevant for the downstream task, as opposed to aspects which are spurious or irrelevant. The price we pay for this is that, unlike measures such as CKA, model stitching depends on the downstream task. However, our results indicate that neural networks tend to learn similar representations for a variety of natural tasks.

*Asymmetry:* Our intuition is that some representations are *better* than others. For example, we believe that with more data, neural networks learn better representations. However, by design, such comparisons cannot be demonstrated by representation similarity measures that only measure the distance between two representations. In contrast, we are able to demonstrate that "more is better" using stitching-based measures. Concretely, CKA and other similarity measures are symmetric, while stitching is not: it may be the case that a representation $\phi$ can be stitched in place of a representation $\sigma$, but not vice-versa. Also, stitching a representation $\phi$ can (and sometimes does) *improve* performance.

*Interpretable units:* Measures such as CCA/CKA give a number between 0 and 1 for the distance between representations, but it is hard to interpret what is the difference, for example, between a CKA value of 0.9 and value of 0.8. In contrast, if the loss function $\mathcal{L}$ has meaningful units, then the stitching penalty inherits those, and (for example) a representation $r$ having a penalty of 3% in CIFAR-10 accuracy has an operational meaning: if you replace the first layers of the network with $r$, the decrease in accuracy is at most 3%.

*Invariance:* A representation similarity measure should be invariant to operations that do not modify the "quality" of the representation, but it is not always clear what these operations are. For example, CCA and CKA are invariant under orthogonal linear transformations, and some variants of CCA are also invariant under general invertible linear transformations (see Table 1 in Kornblith et al. [2019]). However, it is unclear if these are the natural families. For example, randomly permuting the order of pixels (either at the input or in the latents) is an orthogonal transform, and thus does not affect the CKA. However, this completely destroys the spatial structure of the input, and intuitively should affect the "representation quality." On the other hand, certain non-orthogonal transforms may still preserve representation quality– for example, the non-invertible transformation of projecting out spurious coordinates. Thus, orthogonal transforms may not be the right invariance class to consider representations. Using stitching we can explicitly ensure invariance under any given family of transformations by adding it to the stitching layer. Concretely, our choice of using a $1 \times 1$ convolutional stitcher yields much weaker invariance than general orthogonal transforms– and thus, model stitching can predict that shuffling pixels leads to a "worse" representation.

# 4 Stitching Connectivity

We first focus on a special case of model-stitching, wherein we stitch two *identically distributed* networks to each other. That is, we train two networks of same architecture, on the same data distribution, but with independent random seeds and independent train samples. This question has been studied before with varied conclusions [Li et al., 2016, Wang et al., 2018]. We find that empirically, two such networks can be stitched to each other at *all layers*, with close to 0 penalty. This is a new empirical property of SGD trained networks, which we term "stitching connectivity."

---

[2]For the former case, this is highlighted in Figure 6 of Nguyen et al. [2021], where it is stated that "Representations within block structure differ across initializations."

Formally, let $A, B$ be two trained networks of identical architectures with $L$ layers, for a task with objective function $\mathcal{L}$. For all $i \in \{0, 1, ..., L\}$, define $S_i$ to be the stitched model where we replace the first $i$ layers of $A$ with those same layers in $B$ (and optimize the stitching layer as usual). That is, $S_i := \arg\min_{S=A_{>i} \circ s \circ B_{\leq i}} \mathcal{L}(S)$. Observe that $S_0 = A$ and $S_L = B$, so the sequence of models $\{S_0, S_1, \ldots, S_L\}$ gives a kind of "path" between models $A$ and $B$. Further, due to our family of stitching layer and network architecture ($1 \times 1$ conv stitchers and conv-nets), the stitching layer $s$ can be folded into the adjacent model. Thus, all models $S_i$ have identical architecture as $A$ and $B$. We say $A$ and $B$ are "stitching-connected" if all the intermediate models $S_i$ have low test loss.

**Definition 1** (*Stitching Connectivity*)**.** Let $A, B$ be two networks of identical architectures. We say $A$ and $B$ are *stitching-connected* if they can be stitched to each other at all layers, with low penalty. That is, if all stitched models $S_i$, defined as above, have test loss comparable to $A$.

Stitching connectivity is not a trivial property: two networks with identical architectures, but very different internal representations, would fail to be stitching connected. Our main claim is that for a fixed data distribution, almost all minima reached by SGD are stitching-connected to each other.

**Conjecture 2** (*Stitching Connectivity of SGD, informal*)**.** *Let $A_1, A_2$ be two independent and identically-trained networks. That is, networks of the same architecture, trained by SGD with independent random seeds and independent train sets from the same distribution. Then, for natural architectures and data-distributions, the trained models $A_1$ and $A_2$ are stitching-connected.*

This conjecture states a structural property of models that are likely to be output by SGD. If we run the identical training procedure twice, we will almost certainly not produce models with identical parameters. However, these two different parameter settings yield essentially equivalent internal representations– this is what it means to be stitching-connected.

**Discussion.** Stitching connectivity is similar in spirit to mode connectivity [Garipov et al., 2018, Freeman and Bruna, 2017, Draxler et al., 2018], in that they are both structural properties of the set of typical SGD minimas. Mode connectivity states that typical minima are connected by a low-test-loss path in parameter space. Stitching connectivity does not technically define a *path* in parameter space– rather, it defines a *sequence* $S_0, S_1, \ldots, S_L$ of low-loss models connecting two endpoint models. For each model in this sequence, all but one layer is identical to one of the endpoint models. Thus, we can informally think of the stitching sequence as a different way of "interpolating" between two models.

The stitching connectivity of SGD is especially interesting for overparameterized models, since it sheds light on the *implicit bias* of SGD. In this case, there are exponentially-many global minima of the train loss, even modulo permutation-symmetry. Apriori, it could be the case that each of these minima compute the classification decision in different ways (e.g. by memorizing a particular train set). However, empirically we find that SGD is "biased" towards minima with essentially the same internal representations– in that typical minima can be stitched to each other.

**Experiments.** Figure 2A demonstrates the stitching-connectivity of SGD via the following experiment. We train a variety of model architectures on CIFAR-10, with two different randomly initialized models per architecture. We consider a ResNet-18, two variants of ResNet-18 with $0.5\times$ and $2\times$ width, a significantly deeper ResNet-164, a feed-forward convolutional network Myrtle-CNN [Page, 2018] and a Vision Transformer [Dosovitskiy et al., 2020] pretrained on CIFAR-5m [Nakkiran et al., 2021]. We stitch the two randomly initialized models at various intermediate layers, forming the stitching sequence $S_0, \ldots S_L$. Figure 2A plots the test errors of these intermediate stitched models $S_i$ with the first and last point showing the errors of the base models. For all layers $i$, the test error of the stitched model $S_i$ is close to the error of the base models. A similar result holds for networks trained on disjoint train sets. Full experimental details are provided in Appendix A.

## 5    All Roads Lead to Rome

In this section, we quantify the intuition that "all roads lead to Rome" in representation learning: many diverse choices of train method, label quality, and objective function all lead to similar representations in early layers. However, we find that such training details can affect the representation at later layers.

**Comparing self-supervised and supervised methods.** If we train models with the same architecture and train set, but alter the training method significantly, what do we expect from the representations? To explore this, we compare the representations of two very different training methods — standard

end-to-end supervised training (E2E) versus self-supervised + simple classifiers (SSS), that first learn a representation from unlabeled data and then train a simple linear classifier on this representation.[3]

SSS algorithms like SimCLR [Chen et al., 2020b], SwAV [Caron et al., 2021a] and DINO [Caron et al., 2021b] etc have recently emerged as prominent paradigm for training neural networks that achieve comparable accuracy to E2E networks. We stitch the representations of these SSS algorithms to an E2E supervised network (all with a ResNet-50 backbone) trained on ImageNet. While all of these methods achieve similar test accuracy on a ResNet-50 backbone of $75\% \pm 1$ (except SimCLR at 68.8%) [Goyal et al., 2021], they are trained very differently. Figure 2B shows that the SSS trained networks are stitching connected at all layers to the E2E network. This suggests that while the advances in SSL have been significant for learning features without labels, the features themselves are similar in both cases. This is in agreement with prior work which shows that SSS and E2E networks have similar texture-shape bias and make similar errors [Geirhos et al., 2020]. Since certain SSS algorithms have been proven to have small generalization gap [Bansal et al., 2021], the similarity of representations between SSS and E2E algorithms may yield some clues into the generalization mystery of E2E algorithms. A similar experiment for SimCLR with ResNet-18 trained on CIFAR-10 is shown in Appendix B.2, along with the CKA for the same networks. We find that the CKA varies between $0.35 - 0.9$ for different layers, while stitching gets maximum stitching penalty of 3%.

**Changing the label distribution $p(y|x)$:** How much does the representation quality depend on label quality? To explore this, we compare the representations of networks with the same input distribution $p(x)$, but different label distributions $p(y|x)$. We take the CIFAR-10 distribution on inputs $p(x)$, and consider several "less informative" label distributions: (1) Coarse labels: We super-class CIFAR-10 classes into a binary task, of Objects (Ship, Truck, etc.) vs. Animals (Cat, Dog, etc.) (2) Label noise: We set $p = \{0.1, 0.5, 1.\}$ fraction of labels to random labels. We then stitch these "weak" networks to a standard CIFAR-10 network at varying layers, and measure the stitching penalty incurred in Figure 3A. We find that even with poor label quality, the first half of layers in the "weak" model are "as good as" layers in the standard model (with the exception of the weak network trained on 100% noisy labels). These experiments align with the results of Nakkiran and Bansal [2020], which show that neural networks can be sensitive to aspects of the input distribution that are not explicitly encoded in the labels. Full experimental details appear in Appendix A.

These results are in agreement with prior work on vision [Olah et al., 2017], which suggests that the first few layers of a neural network learn general purpose features (such as curve detectors) that are likely to be useful a large variety of tasks. Formalizing the set of pre-training tasks for which such similar representations are learnt is an important direction for future work to understand pretraining.

# 6  More is Better

Now, we use stitching to quantify the intuition that "more is better" for representations. That is, larger sample size, model size, or train time lead to progressively better representations *of the same type*.

**Number of samples:** When scaled appropriately, neural network performance improves predictably with the number of training samples (e.g. [Kaplan et al., 2020]). But how does this improvement manifest in their representations? We investigate this by stitching the lower parts of models trained $\{5K, 10K, 25K\}$ samples of CIFAR-10 to the upper layers of a model trained with $10K$ samples.

First, as Figure 2C shows, we observe that the models are all stitching compatible— the *better* representation trained with $25K$ samples can simply be "plugged in" to the model trained with fewer samples, and this *improves* the performance of the stitched network relative to the $10K$ network. Note that there is no theoretical reason to expect this compatibility, better networks could have learnt fundamentally different features from their weaker counterparts. For example, a network trained on few samples could have learnt only "simple features" (presence of sky in the image / presence of horizontal lines), while one trained on many samples may learn only "complex features" (presence of an blue eye / presence of a furry ear), which are not decodable by the weaker network.

Secondly, we find that some layers are more *data hungry* than others. When stitched at the first few layers, all the models have similar performance, but the performance degrades rapidly with fewer samples in the mid-layers of the network. This suggests that each layer of a neural network has its

---

[3]We use this approach rather than the more standard SSL approach of self-supervised training with fine tuning to ensure that all layers except the last were learned without the use of labels.

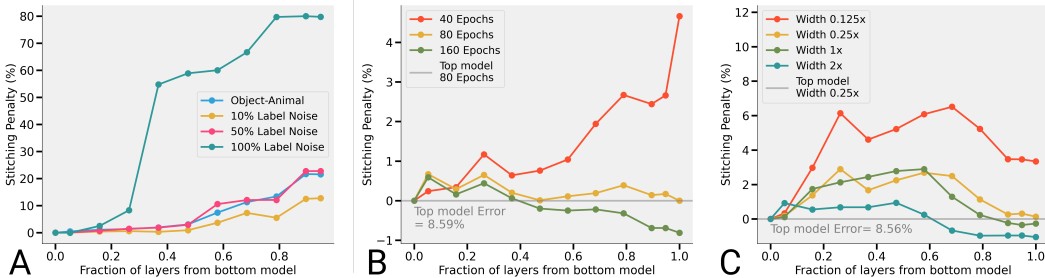

Figure 3: (A) **Changing the label distribution:** Representations trained on CIFAR-10 with the Object vs. Animals task or with {10%, 50%, 100%} label noise and stitched to a network trained on original CIFAR-10 labels. Early layers learn similar representations even when the label distribution is "less informative" than training with all labels (B) **Increasing training time:** Representations at different epochs during training are stitching compatible and early layer converge faster (B) **Increasing width:** Better representations from a wider network can be stitched with a thinner network to improve performance. All experiments were performed with ResNet-18 on CIFAR-10.

own *sample complexity*. As a corollary of this finding, we predict that we can train some of the layers with few samples, freeze them and train the rest of the model with larger number of samples. Indeed, we find that we can recover most of the accuracy of the network by training the first three, and the last three layers of this model (about half of the layers) with just $5K$ samples, freezing them and training the rest of the model with all the samples to obtain a network within $2\%$ accuracy of the original network. See Appendix B.4 for details of the experiment and training plots. This is similar to the "freeze-training" experiments suggested by Raghu et al. [2017].

**Training time:** We observe similar results when we stitch networks trained for different number of epochs. We train a ResNet-18 trained on CIFAR-10 and stitch the representations from the model at $\{40, 80, 160\}$ epochs to the model at the $80$-th epoch. As training time increases, Figure 3B shows that the representations improve in a manner that is compatible with the earlier training times. Note that this is a nontrivial statement about neural network training dynamics: It could have been the case that, once a network is trained for very long, its representations move "far away" from its representations near initialization – and thus, stitching would fail. However, we find that the representations at the end of training remain compatible with those in the early stage of training. We also observe that earlier layers converge faster with time, as was shown by Raghu et al. [2017].

**Width:** Similarly, we train models with the same architecture but varying width multipliers $\{0.25\times, 1\times, 2\times\}$ (Figure 3C). We find that "better" models with higher width models can be stitched to those with lower width and improve performance, but not vice versa. The CKA between representations with $0.25\times$ and $2\times$ width multiplier is in the range $0.7 - 0.9$ (See Appendix B.2)

Taken together, these results suggest that neural networks obey a certain kind of *modularity* — better layers can be plugged in without needing to re-train the whole network from scratch.

## 7 Conclusion and Future Work

As our work demonstrates, model stitching can be a very useful tool to compare representations in interpretable units. Model stitching does have its limitations: it requires training a network, making it more expensive in computation than other measures such as CKA. Additionally, stitching representations from two different architectures can be tricky and requires a careful choice of the stitching family. While we restricted ourselves to stitching the first $l$ layers of a network, stitching can be used to also plug in intermediate layers or parts of layers, and in general to "assemble" a new model from a collection of pre-trained components.

There are various avenues for future research. All of our results are in the vision domain, but it would be interesting to compare representations in natural language processing, since language tasks tend to have more variety than vision tasks. It would also be interesting to study the representations of adversarially trained networks to diagnose why they lose performance in comparison to standard training. In general, we hope that model stitching will become a part of the standard diagnostic

repertoire of the deep learning community. **Societal impacts:** This paper makes methodological and foundational contributions that do not have direct impact on society. Model stitching can potentially be used to understand and develop better representation learning mechanisms. While this could indirectly lead to future applications, it is premature to predict their positive or negative impacts.

## 8    Acknowledgements

YB is supported by IBM Global University awards program and NSF Awards IIS 1409097. PN is supported in part by a Google PhD Fellowship, the Simons Investigator Awards of Boaz Barak and Madhu Sudan, and NSF Awards under grants CCF 1565264, CCF 1715187. BB is supported by NSF award CCF 1565264, a Simons Investigator Fellowship and DARPA grant W911NF2010021. We thank MIT-IBM Watson AI Lab and John Cohn for providing access and support for the Satori compute cluster.

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
