# A  Experimental Details

## A.1  Networks used for comparison

## A.2  CIFAR-10:

**ResNets:** We train a variety of ResNets for comparing representations. The base ResNet architecture for all our experiments is ResNet-18 [He et al., 2015] adapted to CIFAR-10 dimensions with $64$ filters in the first convolutional layer. We also train a wider ResNet-w2x and narrower ResNet-0.5x with $128$ and $32$ filters in the first layer respectively. For the deep ResNet, we train a ResNet-164 [He et al., 2015].

For the experiments with varying number of samples or training epochs, we train the base ResNet-18 with the specified number of samples and epochs.

For experiments with changing label distribution, we also train the base ResNet-18. As specified, for the binary Object-Animal labels, we group `Cat`, `Dog`, `Frog`, `Horse`, `Deer` as label $0$ and `Truck`, `Ship`, `Airplane`, `Automobile` as label $1$. For the label noise experiments, with probability $p = \{0.1, 0.5, 1.0\}$ we assign a random label to $p$ fraction of the training set.

All the ResNets are trained for $64K$ gradient steps of SGD with $0.9$ momentum, a starting learning rate of $0.05$ with a drop by a factor of $0.2$ at iterations $\{32K, 48K\}$. We use a weight decay of $0.0001$. We use the standard data augmentation of `RandomCrop(32, padding=4)` and `RandomHorizontalFlip`.

**Vision Transformer:** We use a ViT model [Dosovitskiy et al., 2020] from the `timm` **?** library adapted for CIFAR-10 image dimensions. The model has patch size $= 4$, depth $= 12$, number of attention heads $= 12$ and dimension $= 768$. We train two models with different initializations on CIFAR-5m [Nakkiran et al., 2021], which consists of 5 million images synthetic CIFAR-10 like images generated from a denoising diffusion model. We perform the stitching on CIFAR-10.

**Myrtle CNN:** We consider a simple family of 5-layer CNNs, with four Conv-BatchNorm- ReLU-MaxPool layers and a fully-connected output layer following Page [2018]. We train it for $80K$ gradient steps with a constant learning rate of $1$. We use the standard data augmentation of `RandomCrop(32, padding=4)` and `RandomHorizontalFlip`.

## A.3  ImageNet:

For the supervised model, we use the pretrained model from PyTorch. For the SwAV and SimCLR models, we use the pretrained models provided by Goyal et al. [2021]. For the DINO model, we use the pretrained model provided by Caron et al. [2021b].

## A.4  Stitcher

**Convolutional networks:** In all our stitching experiment, the top consists of a convolutional network and the representation $r$ comes from a convolutional network $B$. For most experiments, the architecture of $B$ is the same as $A$, with the except of the "more width" experiments where it may have different width but the same depth. Let's say the the first $l$ layers of the bottom model $B$ have channels $C_1$ and the top model $A$ has channels $C_2$. The stitching layer is then { `BatchNorm2D`$(C_1)$`,` `Conv(in features = ` $C_1$`, out features = ` $C_2$`, kernel size = 1)`, `BatchNorm2D`$(C_2)$ `}`.

For computation reasons, in the residual networks, we perform the stitching between two ResNet blocks (not inside a residual block). There is no reason to expect different results within the block.

**Vision Transformer:** We use a linear transform of the embedding dimension ($768 \times 768$).

All stitching layers were optimized with Adam cosine learning rate schedule and initial learning rate $0.001$

# B   Additional results

## B.1   Ablations

We now perform ablations on the stitching family $\mathcal{S}$ for convolutional networks. We set the kernel size of the stitching convolutional layer to $\{1, 3, 5, 7, 9\}$ and check how this affects the test performance. Both the top and bottom models are ResNet-18 trained on CIFAR-10 with different random initializations. We find that kernel size has minimal impact on the test error of the stitched network. We choose a kernel size of 1, so that the stitched model architecture is identical to either the top or the bottom model.

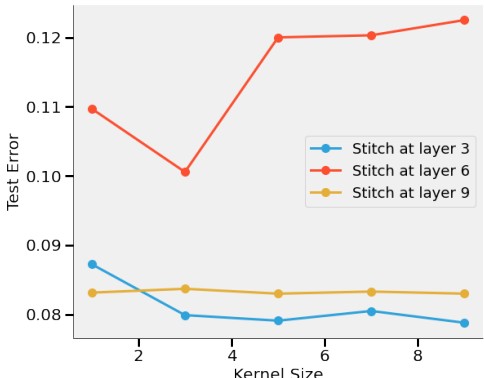

Figure 4: Test Error with changing kernel size

## B.2   Comparison with CKA

### B.2.1   Random network

To confirm that our stitching layer is not performing learning, we stitch a randomly initialized untrained network to a trained top model. The network architecture is ResNet-18 and the top is trained on CIFAR-10. The results of this experiment for networks trained on CIFAR-10 and ImageNet are shown in Figure 2A and B respectively (also plotted for CIFAR-10 in Figure 5 for clarity). These plots show that the early layers of the model have *low* stitching penalty, implying that the early layers behave similarly to a randomly initialized network. To confirm that this is not a pathological scenario for model stitching, we also compute the CKA at each layer for the same random and top networks. Figure 5 shows that CKA also predicts that the early layers of a neural network are 'similar' to a random network.

### B.2.2   Self-supervised vs. supervised

We now compare the representations a network trained with self-supervised learning (SimCLR Chen et al. [2020b]) or end-to-end (E2E) supervised learning. As Figure 6B shows, the two networks have similar representations - the networks are stitching connected in both directions (SimCLR at the bottom, E2E at the top and vice-versa). Moreover, the stitching penalty is in the same range as that of a different E2E network with a different random initialization. On the other hand, CKA shows that the SimCLR representations can differ a fair bit from the E2E networks. The stitching results show that these representations differ only superficially in directions that are not relevant to the downstream classification performance.

### B.2.3   More is better

We now show CKA comparisons for experiments in Section 6. The results are shown in Figure 7.

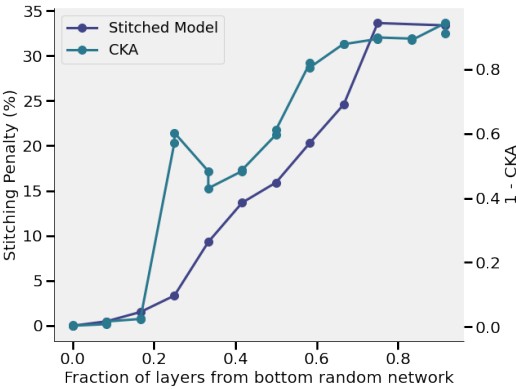

Figure 5: Comparing the representation of a random network with a trained network with model stitching and CKA

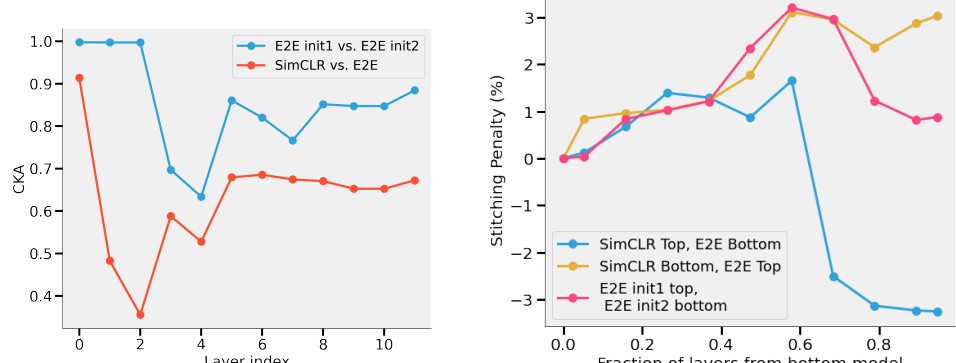

Figure 6: Comparing representations for a SimCLR trained network with a fully supervised network on CIFAR-10 (a) With CKA. CKA shows that representations trained with SimCLR vs. E2E differ more than those from two E2E networks with different random initializations (b) With Stitching: Stitching shows that the SimCLR and E2E representations are identical in stitching performance

**Width:** We compare the CKA for all layers for a ResNet-18-w0.25x with ResNet-18-w2x. We find that the CKA can be as low as $0.7$ for the early layers, while the stitching penalty for early layers is small as shown in Figure 3C.

**Samples:** We compare the CKA for a network trained with $5K$ samples with a network trained with $25K$ samples. The CKA can go as low as $0$. On the other hand, stitching in Figure 2C shows a negative stitching penalty, showing that the layers trained with more samples are still *compatible* with the $5K$ model.

**Training time:** We compare the CKA for a network at the end of training (160 epochs) with the midpoint of training (80 epochs). CKA finds that the representations are similar.

## B.3 Comparison with fine-tuning

We now show that finetuning can overestimate the similarity between representations compared to model stitching. To do so, we compare the representations of a trained ResNet-164 (top model) with a randomly initialized untrained ResNet-164 (bottom model). For finetuning, we simple freeze the first $l$ layers of the bottom network and train the top layers after reinitializing them. The results are shown in Figure 8.

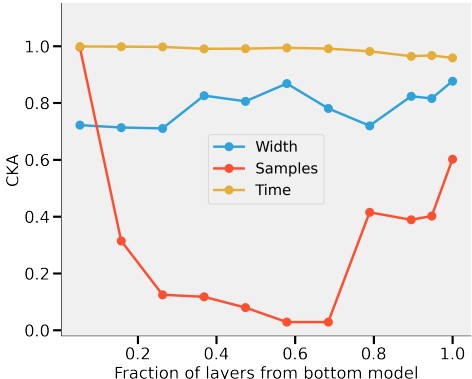

Figure 7: Comparing representations with CKA in settings where one model has better accuracy than the other obtained by (1) Increasing width (2) Increasing training time (3) Increasing the number of samples. We find that CKA gives mixed answers about the similarity of such representations, but model stitching shows that 'better' representations can be stitched into a weaker model to gain performance.

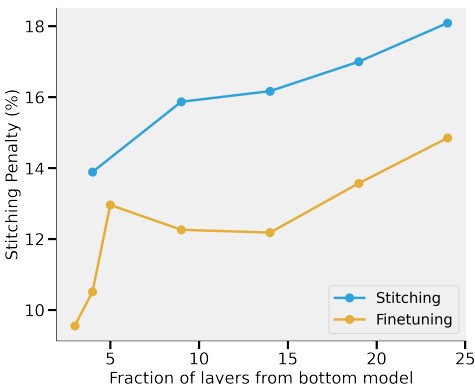

Figure 8: Finetuning vs. Stitching: We compare the effect of fine-tuning (the top model) vs. just stitching the stitching layer. We find that fine-tuning can over-estimate the similarity since fine-tuning has many more trainable parameters.

## B.4 Freeze training with fewer samples

The results in 6 suggest that certain layers of the network have smaller *sample complexity* than the other layers. That is, they take fewer samples to reach a representation that is stitching-connected to the rest of the model. To test this hypothesis further, we take a ResNet-18 trained on $5K$ samples ($10\%$) of the dataset and stitch it with a top network trained on the full dataset (results in Figure 2C) and the reverse - a top model with $5K$ and bottom with $50K$. We then choose the layers whose stitching penalty is close to 0 at $5K$ (layers $\{0, 1\}$ in Figure 2C) and similarly for the opposite (layers $\{8, 9, 11\}$). Then, we freeze these 5 layers and train the rest of the network with the full dataset. The test curves with training time are shown in Figure 9. As we can see, this network obtains good performance (up to $\approx 3\%$). This suggests that not all layers need to be trained for a large number of samples - this can potentially be used in future applications to speed up training time. Predicting which layers will have small sample complexity is an interesting direction for future research.

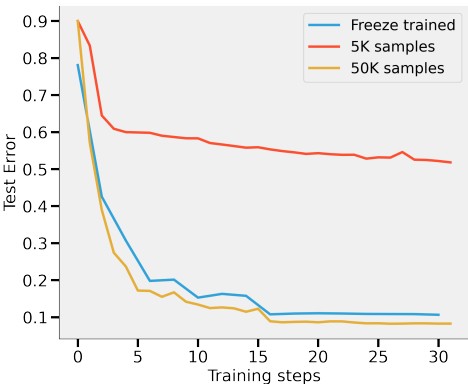

Figure 9: Freeze training