# OpenReview forum: "Revisiting Model Stitching to Compare Neural Representations"
_NeurIPS.cc/2021/Conference — NeurIPS 2021 Poster_

### Official Review · Reviewer_myvx · 2021-07-11

**Rating:** 7
**Confidence:** 3

**Summary:**

The paper use model stitching as a way to examine the hidden states of neural network models. The proposed method stitches the top and bottom of two similar models together with a linear transformation (if you don't take BatchNorm into account). Experiment results show that the hidden states, learned from different initializations, different subsets of the training set and even different losses can be translated from one to another. Such that the top of one model can use the hidden state given by another model to predict the same label as the original top model. Based on this observation the paper proposed stitching connectivity, a measurement of similarity between semantic space of hidden states.

**Limitations And Societal Impact:**

1. Like most theoretical papers, the conclusion of this paper is based on two assumptions:
a) linear transformation with BatchNorm is not a strong translation mechanism that can map one type of representation to another;
b) the inputs (hidden states), that provide similar outputs (labels), should share a similar representation space.
Although the two assumptions seem intuitively correct, more theoretical or empirical evidence will further strengthen the paper.

2. It would also be interesting to examine what information or features are preserved through the stitch linear. Since the author argued that the stitching layer can ignore spurious features.

3. Line 256-259: All experiments are done with the ResNet model and Adam optimizer, but the author generalized their conjecture to SGD and neural network models. It seems overclaimed.

4. The related works section is not comprehensive, it didn't include other works that study similar questions.


**Main Review:**

Overall the paper is interesting and convincing. It studies a long-standing question of deep learning -- whether the hidden states of different models are modeling the same features.
The idea of model stitching is already proposed in previous work (Lenc & Vedaldi 2015). But the paper proposes to use the stitching penalty as a quantitative metric to study the question.
The experiments are comprehensive. The author proposed several experiment settings to show the advantages of the "stitching penalty" over other metrics. Experiment results also show that good networks of the same architecture, but trained in very different ways can be stitched to each other without a drop in performance.
The conclusion should be interesting to a wide spectrum of researchers, for example, whether pre-trained language models learned the same set features.
The paper is well-written. Although the paper is not organized in a regular way, presenting detailed results in the introduction actually makes the paper easier to read.

**Time Spent Reviewing:**

5

---

> ### Author Response · Authors · 2021-08-10
> **Response to reviewer myvx**
>
> Thank you for your review. We agree that the paper is based on the assumption that linear transformation with BatchNorm is not strong enough to learn arbitrary maps. While it would be difficult to verify this theoretically, we try to validate this by stitching with a random network. If the stitcher was too strong, then the random network would also be able to achieve a low stitching penalty at all layers, which is not the case.
>
> Studying which information is preserved through stitching is indeed an interesting question! If we can find a way to ‘isolate’ which directions are being ignored by the stitcher, we could find a more compact representation.
>
> Regarding the generalization of results from Adam to SGD:
> --- SGD or Adam for the trained top/bottom models. The reason for our claim that the choice of optimizer doesn’t impact the result is that some networks in our experiments were trained with SGD (ResNets) and some with Adam (ViT) and this did not have an impact.
> --- SGD or Adam for the stitcher: Since the stitcher is under-parameterized (a fact also demonstrated by its failure to boost performance for random networks), we expect the low-penalty stitcher to be uniquely determined and hence independent of the choice of optimization method used to find it..
>
> We tried to incorporate related works in the body of the paper instead of a separate section so that we could discuss the papers in context. We are happy to add an additional Related Works section with more references in the next revision.

---

### Official Review · Reviewer_tKuq · 2021-07-13

**Rating:** 7
**Confidence:** 4

**Summary:**

Understanding how the representations of neural networks, trained in diverse settings, compare and can be combined is an important problem in representation learning. This papers addresses this problem by studying model stitching (Lenc & Vedaldi 2015) in the following experimental setting: connect the bottom-layers of A to the top-layers of B with an inexpensive trainable layer, where A and B are trained neural networks. The contributions are:

1) Empirical confirmation of inutitive statements: such as:

* representations trained with more data and training time or with larger width can be stitched to representations trained with less resources;

* representations trained with different objectives (supervised vs. self-supervised) can be stitched together with minimal penalty;

* early layers of representations are similar even when the labels used for training are different.

2) Observation of a new structural property of SGD, defined as "stitching connectivity" by the authors: typical minima reched by SGD can be stiched together without changing accuracy much.







**Limitations And Societal Impact:**

I think the big thing that the paper should discuss is why the stitching behavior happens for natural datasets. I imagine, given a fixed neural network architecture, one could design a dataset, where stitching connectivity would fail. In fact, it could be possible to optimize for such a dataset with meta learning. Any thoughts about that?

**Main Review:**

# Originality

The idea to revisit model stitching is timely. While intuitive, the experiments are cleverly designed, and demonstrate the benefits from model stitching.

# Quality

The study is methodical and useful. While reading the paper, most of my questions were eventually answered in later parts of the text.

# Clarity

The paper is overall clear, well-organized and engaging. Some suggestions for improvement:

* Line 103: Garipov et al. is missing a year.

* Line 122: simple of case <- simple case

* Line 133 is ambiguous: “lies in \mathcal{A}” is not very formal, as you haven’t defined what \mathcal{A} is formally. Line 113*  doesn’t help. Why is the composition part of \mathcal{A}. You answer my question in lines 246-248, but I think you should make this comment earlier.

* Table 1: What does \textbf{Better} mean here? “Far” for CKA means representation similarity, but “Better” means that stitching improves the method. These comparisons are different. You address my question in line 189, but I think you should note it in the Table’s caption.

* Line 212-214: Have you studied this possibility? It might be interesting.

* Line 249: “low test loss” is not a formal definition. Does that affect your discussion?

* Line 342-351: this is an interesting experiment, but it may not be very realistic. If the labels are available, why not use them to update the first and last layers too? Apart from reducing computation for training, is there any situation where freezing the first and last layers would be better?

* Lines 78-79: “much better suited tool” <- I would suggest to rephrase this, because studying representations should be more specified. Depending on what one wants to study, different tools might be preferred. As the conclusion section of [Kornblith et al., 2019] clarifies, measuring similarity between representations is an “ill-defined problem.”

* What is the legend of the colors in Figure 4 (in SM). It's not labeled currently.

* Figure 6 (in SM): in the caption, "differ more than two" seems to be missing a word. Also, the captions talks about (a) and (b), but we do not see it in the Figure. SimCLR and E2E do not seem "identical."


# Significance

The convincing empirical results about model stitching would likely aid researchers and practitioners. The former can study interesting follow ups from this study, such as the "sample complexity of individual layer" and the later can improve applications, such as stitching two high-performing, but complimentary, neural networks.





**Time Spent Reviewing:**

5

---

> ### Author Response · Authors · 2021-08-10
> **Response to Reviewer tKuq**
>
> Thank you for your review, and the detailed comments regarding clarity. We will fix all of these in the next revision of the paper. Some specific responses are:
> (1) Line 212-214: Yes we did, Figure 1C shows that performance is improved if we stitch a network trained with more samples to a top trained with fewer samples
> (2) Line 249: Yes, it is not formal. In general, we mean that the difference is comparable to than the variance that would be observed for identical networks but trained with different random seeds.
>
> We agree that it is an interesting question to understand why stitching occurs for natural datasets and for SGD trained networks!  We did try to construct two networks of similar performance on CIFAR-10 that did not have stitching connectivity by trying unique optimization schedules (freezing some layers and only training them later) or adding random transforms and their inverse between some layers. However, we could not find any cases where stitching failed in our limited efforts.  Constructing a different dataset where this occurs is a great suggestion - thanks!

---

### Official Review · Reviewer_CLM9 · 2021-07-14

**Rating:** 7
**Confidence:** 4

**Summary:**

The authors studied model stitching as a methodology to examine the internal representations of neural networks. Particularly, they use model stitching to verify several statements such as “good networks learn similar representations” and “representations learned with (1) more data, (2) bigger width, or (3) more training time can be “plugged in” to weaker models to improve performance". In addition, They showed that SGD has a structural property akin to mode-connectivity: typical minima reached by layers trained via SGD can all be stitched to each other with minimal change in accuracy, termed as  “stitching connectivity”.


**Limitations And Societal Impact:**

The authors note that there’s no obviously foreseeable negative societal impact of the work, which I largely agree with.

**Main Review:**

The work is only marginally innovative since the idea of model stitching was introduced a long time ago by Lenc and Vedaldi (2015) and the current work only extends the method to make it applicable in new contexts. It’s nice that the work shows that model stitching is more powerful than originally conceived, and it has several benefits over representation-similarity measures, such as ignoring spurious features, asymmetry, interpretable units, and invariance. However, the innovation is still incremental.

 The submission is relatively sound. The authors verify several intuitive statements such as “good networks learn similar representations” and “more is better” with the model stitching approach. I do have a couple of concerns. In section 4 Stitching connectivity, the author mentioned “We find that empirically, two such networks can be stitched to each other at all layers, with close to 0 penalty. This is a new empirical property of SGD trained networks, which we term “stitching connectivity.” However, the conclusion is based on experiments with a limited set of computer vision models that are primarily convolutional networks on a single dataset CIFAR-10. I doubt that any general argument about the empirical property of all SGD trained networks,  i.e., “stitching connectivity.” could be derived from these limited experiments.

My second concern is about line 318: “We find that even with poor label quality, the first half of layers in the “weak” model are “as good as” layers in the standard model. Even when the model is given less informative labels, such as coarse labels (object vs. animal) the first few layers in the weak network learn representations as good as the “strong” networks". I think this is very counter-intuitive and perhaps very experiment-specific. It’s likely due to the fact that the original CIFAR-10 dataset only has ten classes and reducing 10 classes to 2 classes does not make a big difference. I wonder if the authors have tested the model on a dataset having much more classes like ImageNet. If reducing the class numbers still leads to near-zero stitching penalties in the first few layers in the “weak” neural network, then the author’s argument that “the first few layers of a neural network learn general purpose features (such as curve detectors) ” (line 324) would be more convincing.

My third question is related to the training of the stitcher. In Appendix Line 565, the authors give a brief description of the procedure of training the stitcher. I’m curious to know the number of epochs for the stitcher training (if fixed)  or any strategies for early-stopping if used. This is important because the stopping criteria should be the same for training the stitcher and training the original model. Otherwise, the performances can not be fairly compared and the stitching penalty cannot be reliably obtained.

The submission is clearly written and well organized. If the authors can address the issues I mentioned above and make the arguments and conclusions in the paper more convincing, the results can be important in the sense that it shows model stitching can be a powerful tool to study representation learned by deep nets. Particularly, this method can be used to compare representation similarity, determine which representation is better and understand the impact of different representations on model performance via stitching penalty.  Future studies could use this approach to understand deep neural nets better.

-----

Update: I've read the authors' responses to my review as well as the ones to the other reviewers'. I think their responses addressed some of my questions. I appreciate the clarification on the training procedure. I still think that the authors need to make it clear that the property observed for the SGD trained networks in this paper are purely empirical and whether it generalizes beyond the compute vision models requires additional experiments in future research. Given the improvements and the limitations, I decided to raise my score to 7 but no further.


**Time Spent Reviewing:**

10 hours

---

> ### Author Response · Authors · 2021-08-10
> **Response to Reviewer CLM9**
>
>
> Thank you for your review. While indeed the stitching method was already proposed in 2015, it has not been significantly studied since then. We believe that this is largely because researchers did not realize its potential for studying representations beyond what can be achieved by other methods. In addition to demonstrating the potential of stitching, our results also quantitatively characterize a variety of interesting behaviors of neural networks - such as the similarity of networks trained with supervised and self-supervised learning - beyond what was known in prior literature..
>
> Regarding the generality of our results, we would like to emphasize that we also experimented with Vision Transformers (Figure 1A), and we have results for two networks trained on ImageNet with different random initializations (the result looks similar to Figure 1B, and we will add that to the plot as an additional reference line in the next revision). As mentioned in the common response, we also conducted experiments stitching networks of different depths that we will include in the revised version. We agree that it would be interesting to repeat “stitching connectivity” experiments for other domains like language to understand if it is indeed a general property of SGD trained networks, as we also mention in our Conclusion section (Section 7). We would also like to mention that it is common practice in ML literature to highlight phenomenon for a single domain (eg: Garipov 2018) for it to be verified later in other domains.
>
> We thank the reviewer for the suggestion of trying coarse-label training on the ImageNet dataset as well. As mentioned in the common response, the space of possible experiments for stitching is very large, but this is an excellent idea for an experiment, and we will  include it in the next iteration of the paper. We also note that our arguments for the “general purposeness” of early layers are supported not just by the weak label results but also by our self-supervised stitching, which were conducted on ImageNet.
>
> We note that the stitcher’s layer low capacity prevents it from being too dependent on training epochs and early stopping, since (unlike the full network) it is not over-parameterized and so the minimum that the training method ends up in should be unique. That said, we do use the same number of epochs as the original network and do not use early stopping (we did not observe any overfitting either). The low capacity of the stitching layer is also demonstrated by the fact that it fails to induce a significant performance boost when stitching a random (i.e., untrained) network.

---

### Official Review · Reviewer_Anx8 · 2021-07-16

**Rating:** 5
**Confidence:** 4

**Summary:**

The work proposes to utilize model stitching as a tool to compare the quality of learned neural representations. Authors show that model stitching based comparison can avoid many problems often observed in other representation comparison methods, such as CCA/CKA. Using this tool, under image classification setting, authors empirically show that (1) networks with different random initializations would converge to "similar" representations, (2) networks trained via different methods also have "similar" representations given these networks have close performances, and (3) networks trained with more data consistently have better quality.

**Limitations And Societal Impact:**

For limitation, see the comments above. I don't see any concern w.r.t. negative societal impact.

**Main Review:**

The problem of comparing learned neural representations is definitely an important one. And the solution proposed in this work is well motivated and can provide more accurate and richer information of the representation compared to other methods. The conclusions made in the work are also very intriguing.

The main limitation of the current study is the stitching is only performed on a pair of models with exactly the same architecture. If the tool could only be used in this way, its value would be quite limited and the corresponding conclusions would become less interesting.

I was wondering why authors didn't try to stitch models of different architectures.
- One simplest case is to stitch two models with similar computation cost but different hyper-parameters from the same architecture family, for example a ResNet-18 and a ResNet-50 with 0.5x width. Effectively, this would provide meaningful information about the depth/width trade-off. Also, the experiment should be easy to conduct.
- More general case would be stitching models from different architecture families. This test is also important in practice, as people often hope to compare representations from different architecture families. If model stitching does not provide meaningful signal here, this would significantly restrict its value.

Another limitation also acknowledged by the authors is that all experiments are done in image classification. Hence, it remains unclear whether the method is general enough for other problems (say language pretraining where lots of pretrained models have been open sourced).

A minor issue is the stitching penalty is originally defined as the loss gap (Line 120). But in Figure 1, percentages are used to present the stitching penalty, which is inconsistent.

In summary, I feel the work is targeting an important problem and has proposed a well motivated method. However, the empirical study is relatively weak, making it difficult to judge the generalness of the proposed method and hence its practical value.


**Time Spent Reviewing:**

3

---

> ### Author Response · Authors · 2021-08-10
> **Response to Reviewer Anx8**
>
> Thank you for your review. We appreciate that you found our work "well motivated" and "very intriguing." We have responded to your specific concerns below. If all your concerns are addressed, we kindly ask that you consider increasing your score to "accept." We are also happy to follow-up if any concerns remain.
>
> We wish to emphasize that our experiments are fairly extensive, and include several representational comparisons which have not been done in prior works. For example, as far as we know we are the first paper to compare representations between self-supervised and fully-supervised learning, as well as the first to consider the effect of varying the sample size.
>
> While there is no fundamental reason stitching would not work between different architecture families, our focus on “stitching” identical architectures is partially because in this setting there is a natural choice of which neurons to stitch to one another, while in radically different architectures (e.g., transformers vs CNNs) similar representations could be spread out across different layers. Prior work in representational similarity (for eg: CKA) has also primarily compared representations within the same architecture family.
>
> However, we do go beyond stitching exactly identical architectures, such as stitching networks of different widths in Figure 3C. We also conducted experiments stitching different depths (ResNet-20 with ResNet-110) that we did not include in our submission due to page limitations, but we will include them in the final revision. In summary, for each layer in ResNet-20, there is a corresponding layer in ResNet-110 that has a low stitching penalty. However, these layers may occur at different depths in ResNet-110.
>
> That said, we completely agree with the reviewer that it will be interesting and informative to carry out many more experiments, including stitching different architectures and on different domains. We identified all these limitations ourselves in the Conclusion section (Section 7). (See also the Common response posted above) However, we believe the existing experimental results already bring significant value to the representation theory community, because
>    (1) They reveal aspects of representations and training dynamics which were not known before (e.g. "stitching connectivity").
>    (2) They clarify properties of representations which were ambiguous or inconsistent in prior works (e.g. prior works had inconsistent conclusions about whether two networks with fresh random initializations had "similar representations" or not [Li et al., 2016, Wang et al., 2018]).
>
> We were surprised that the stitching method hasn’t really been picked up for deeper exploration of representations since Lenc & Vadaldi’s 2015 paper and hope that our work will encourage more teams to do so.

---

### Author Response · Authors · 2021-08-10
**Common response**

We thank the reviewers for their insightful comments. We are happy that all reviewers agree that we establish stitching as a useful tool for studying neural representations and find our experiments interesting and well-motivated. The main concerns raised are related to doing more experiments in a wider variety of settings (for other domains like language, for more datasets or more architectures). We agree that the space of potential experiments for this method is very large, which is why we find the model of “stitching” so exciting. We chose to focus on this work on settings which (1) are most easily comparable to prior works on similarity measures, (2) have a restricted number of “free parameters” and so the choice of stitching method is relatively constrained, and unlikely to affect the end result, and (3) were feasible within our computational budget. However, we do hope that future work will use stitching to address other questions in representation learning including architectures (depth vs width, transformers vs. CNNs), domains, and more.  Moreover, we did do some experiments (stitching networks of different depth) which we did not previously report in the paper due to page limits, and will add description of those to the revised submission (see response to reviewer Anx8 below).

---

### Decision · Program_Chairs · 2021-09-27

**Decision:**

Accept (Poster)

**Comment:**

This paper explores a previously described technique called "model stitching", in which two trained models can be appended.  Through several empirical experiments the authors show that this technique provides insight into the representational spaces learned by the separate models.

The reviewers thought this idea was interesting, and the paper was creatively written and clear.  The strongest critiques were about the thoroughness of experiments, stating that the authors did not explore the stitching of very diverse architectures or training on very different datasets.  However, I feel the analyses in this paper are extensive, and so I think it is reasonable to leave some of that further experimentation for future work.